# Design and Joint Position Control of Bionic Jumping Leg Driven by Pneumatic Artificial Muscles

**DOI:** 10.3390/mi13060827

**Published:** 2022-05-26

**Authors:** Zhenhao Dai, Jinjun Rao, Zili Xu, Jingtao Lei

**Affiliations:** School of Mechanical and Electrical Engineering and Automation, Shanghai University, Shanghai 200444, China; hughdai@shu.edu.cn (Z.D.); jjrao@shu.edu.cn (J.R.); zilixu@shu.edu.cn (Z.X.)

**Keywords:** bionic leg jumping, active disturbance rejection control, position control, decoupling control

## Abstract

Using the skeletal structure and muscle distribution of the hind limbs of a jumping kangaroo as inspiration, a bionic jumping leg was designed with pneumatic artificial muscles (PAMs) as actuators. Referring to the position of biarticular muscles in kangaroos, we constructed a bionic joint using biarticular and monoarticular muscle arrangements. At the same time, the problem of the joint rotation angle limitations caused by PAM shrinkage was solved, and the range of motion of the bionic joint was improved. Based on the output force model of the PAM, we established a dynamic model of the bionic leg using the Lagrange method. In view of the coupling problem caused by the arrangement of the biarticular muscle, an extended state observer was used for decoupling. The system was decoupled into two single-input and single-output systems, and angle tracking control was carried out using active disturbance rejection control (ADRC). The simulation and experimental results showed that the ADRC algorithm had a better decoupling effect and shorter adjustment time than PID control. The jumping experiments showed that the bionic leg could jump with a horizontal displacement of 320 mm and a vertical displacement of 150 mm.

## 1. Introduction

Jumping allows creatures to cross obstacles several times higher than themselves and avoid risks [1]. Developing a robotic system that can simulate biological structures and functions and extend the motion of traditional robots is an important research goal. Based on an understanding of biological hopping mechanisms, bionic jumping robots simulate the efficient and stable biological hopping process using bionic design principles and improve the hopping ability of robots [2].

Traditional bionic hopping robots adopt motor, hydraulic, pneumatic, and other driving methods [3,4,5,6]. Pneumatic artificial muscles were invented by the American doctor Joseph L. McKibben in the 1950s to assist patients with hand paralysis. Their working principle is very simple: the circumferential stress of a pressurized inner tube is transformed into an axial contraction force by means of a double-helix braided sheath whose geometry corresponds to a network of identical pantographs [7]. Pneumatic artificial muscles change their shape from a given equilibrium position to another equilibrium position under continuous stimulation, just like natural muscle tissue under chemical or electrical stimulation. This phenomenon, which is consistent with Katchalsky’s theory [8] that the first tendency of the material to expand is counterbalanced by retractile forces, distinguishes them from other pneumatic actuators, such as fluid cylinders, because at a constant pressure, the cylinder moves until it comes to a stop, although friction slows it down. Because of this special working principle, their advantages over conventional actuators such as motors and pneumatic pistons stand out:High power/weight ratio: the power/weight ratio of pneumatic McKibben artificial muscles can be 500 W/kg~2 kW/kg, which surpasses the ratio of electric motors that is in the order of 100 W/Kg [9].Flexibility and compliance: non-pressurized PAMs show the same flexibility as a bladder but become rigid and maintain reasonable flexibility when pressurized.Safety and environmental protection: the main driving mechanism of pneumatic artificial muscles is pressurized air or inert gas. Therefore, compared with other electrical, thermal, or chemical equipment, they are safer and more environmentally friendly [10].

The above advantages of pneumatic artificial muscles have gradually been applied to bionic jumping robots and bionic joints. The bionic hopping robots driven by pneumatic artificial muscles that have been developed in the past decade are shown in Table 1. Bionic robots mainly adopt a monoarticular muscle arrangement. A leopard-like jumping robot adopted a lever arrangement of pneumatic artificial muscles acting directly on its joints [11]. Andrikopoulos [12] attached pneumatic artificial muscles to joint pulleys by wire to form bionic joints and conducted joint position control analysis. Since the maximum contraction rate of pneumatic artificial muscles is between 25% and 30%, the motion range of a single-degree-of-freedom joint of pneumatic muscle and the jumping performance of bionic robots are limited. Monoarticular muscles are muscles that act on a single joint, meaning that their contraction only affects the movement of one joint; biarticular muscles span two joints, affecting both joints when the muscle contracts [13]. Jun [14] attached pneumatic artificial muscles to double joints using a slider–crank structure to design a frog-like jumping robot with improved jumping performance. Yamamoto [15] developed a single-push hopping robot that imitated the human musculoskeletal system. The knee joint and hip joint comprised an antagonistic monoarticular muscle arrangement and biarticular muscle arrangement, respectively.

Scholars have conducted several studies on the position control of PAM-driven bionic joints. Hao [19] used active disturbance rejection control to determine the angle of antagonistic pneumatic artificial muscle joints. Compared with PID control, the steady-state error was reduced by 83% and the response time was reduced by 29%. Ugurlu et al. [20,21] used the dissipation theory to design a stable force feedback controller for each PAM unit that could cope with the inherent nonlinearity of PAMs. They used an antagonistic arrangement of PAMs to form joints, incorporating force control of the PAMs and complete tracking control of the joint position. Zhang [22] simplified the pneumatic artificial muscle output force model into a three-element model and used the Kalman filter to actively compensate for the errors in the model. The single-degree-of-freedom platform experiment verified that compared with nonlinear PID control, the tracking error was reduced by more than 96%. Zhu [23] used a neural network compensation PID control algorithm to control the position and stiffness of bionic quadruped robot joints and obtained good position-control results under finite stiffness.

Research into the construction of a single bionic joint and the control of its position using antagonism has yielded good results. However, the fact that PAM shrinkage limits the range of bionic joint rotation remains a problem. Using the skeletal structure and muscle distribution of the hind limb of a jumping kangaroo as a reference, we designed a bionic jumping leg incorporating PAMs as actuators. Considering the limitation of PAM contraction rates, the bionic leg was constructed with a biarticular muscle arrangement and a monoarticular muscle arrangement to improve its joint motion range. The dynamic model of the bionic leg was constructed using the Lagrange method. Considering the problem of joint angle coupling caused by the biarticular muscle of the bionic leg, an extended state observer was used for decoupling, and a decoupling controller based on active disturbance rejection was designed to control the joint angle of the two-degrees-of-freedom bionic leg. We verified the performance of the algorithm using a simulation and experiment; we conducted the jump experiment to investigate the joint angle change based on centroid trajectory planning.

## 2. Design of Bionic Leg Driven by Pneumatic Artificial Muscles

### 2.1. Physiological Structure and Muscle Distribution of Kangaroos

Kangaroos move at high speeds by using their hind legs to jump instead of run, demonstrating the strongest jumping performance of any mammal. Adult kangaroos can jump up to 2 m at a low speed, and at a high speed they can jump a horizontal distance of up to 6~7 m, with a jump speed of up to 60 km/h and a vertical jump height of up to 4 m. The jumping power of a kangaroo is mainly provided by the hind limbs. By dissecting 52 adult kangaroos, Hopwood [24] found that the thigh length accounted for 23% of the total length of the leg, the calf length accounted for 46%, and the foot accounted for 31%. The main muscles of the hind limbs of a kangaroo are shown in Figure 1 [25,26]. The hip extensors include the biceps femoris (BF), the femorococcygeus (FC), and the semitendinosus (ST); the knee extensor muscles include the vastus lateralis (VL), the rectus femoris (RF), and the sartorius (SAR); and the ankle extensor muscles include the medial and lateral heads of the gastrocnemius (GAS), the plantaris (PL), and the flexor digitorum longus (FDL).

### 2.2. Structure Design

We referred to the skeletal structure and muscle distribution of a kangaroo, specifically the location of the gastrocnemius muscle, to determine the location of the antagonistic pneumatic artificial muscles. The two-degree-of-freedom bionic leg driven by monoarticular and biarticular muscles is shown in Figure 2. It contains two rotating joints, the knee joint and the ankle joint, with a sensor fixed on each, and two PAMs. PAM 1, a monoarticular muscle, is articulated with the bionic leg and the knee joint. PAM 2, a biarticular muscle, is articulated with the knee joint and the ankle joint.

### 2.3. Analysis of Bionic Leg Joint

A sketch of the bionic legs driven by PAMs is shown in Figure 3: $l_{s}$ is the length of the tibial connecting rod; $l_{ki}\;\mathrm{and}\;l_{ai}$ represent the distance between the joints and the PAMs; $l_{p1}\;\mathrm{and}\;l_{p2}$ represent the total length of the pneumatic muscles and connectors; and $\theta_{1}\;\mathrm{and}\;\theta_{2}$ represent the angles of the knee and ankle, respectively.

As shown in Figure 3a, the bionic leg incorporated a mixed driving arrangement of monoarticular and biarticular muscles. According to the geometric relationship, the angle of the knee joint and ankle joint is:(1)$$\theta_{1}=\arccos\left(\frac{l_{k1}^{2}+l_{k3}^{2}-l_{p1}^{2}}{2l_{k1}l_{k3}}\right)$$
(2)$$\theta_{2}=\arccos\left(\frac{\left(l_{s}+l_{k2}\cos\theta_{1}\right)\left(l_{x}^{2}+l_{a2}^{2}-l_{p2}^{2}\right)}{2l_{x}^{2}l_{a2}}-\frac{l_{k2}\sin\theta_{1}\sqrt{\left(2l_{a2}^{2}+2l_{p2}^{2}-l_{x}^{2}\right)l_{x}^{2}-{\left(l_{a2}^{2}-l_{p2}^{2}\right)}^{2}}}{2l_{x}^{2}l_{a2}}\right)$$
where $l_{x}=\sqrt{l_{k2}^{2}+l_{s}^{2}+2l_{k2}l_{s}\cos\theta_{1}}$.

For comparison, both joints were constructed with the same parameters, and the pneumatic artificial muscle was arranged in a single joint. The mechanism diagram is shown in Figure 3b. The knee angle was the same as in Equation (1), and the ankle joint angle was:(3)$$\theta_{21}=\arccos\left(\frac{l_{s}^{2}+l_{a2}^{2}-l_{k2}^{2}-l_{p2}^{2}}{2l_{s}l_{a2}}\right)$$

The parameters of the bionic legs are shown in Table 2. Substituting the parameters into Equations (1)–(3), the variation range of the knee and ankle joint angles can be obtained. The bionic leg with only the monoarticular PAM had a knee joint range of 97.6°~129° and an ankle joint range of 70.3°~123.4°. The bionic leg with the biarticular PAMs had a knee joint range of 97.6°~129° and an ankle joint range of 30.5°~123.4°. Compared with the single-joint bionic leg, the ankle joint motion range increased by 39.8°.

## 3. Biomimetic Leg Dynamics

### 3.1. Force Model of Pneumatic Artificial Muscle

The mechanical properties of a PAM are very similar to those of a biological muscle, with the characteristics of nonlinearity and hysteresis. The authors of [28] proposed a static mechanical model of a PAM:(4)$$F\left(\varepsilon ,p\right)=k_{1}\left(p\right)-k_{2}\left(p\right)\varepsilon +k_{3}\left(p\right)\exp\left(-\mu \varepsilon \right)$$
where F(ε,p) is the tension generated by PAM contraction; $k_{i}\left(p\right)$ is the undetermined pressure function; $k_{i}\left(p\right)=k_{i1}+k_{i2}p$, $k_{i1}$, and $k_{i2}$ are the fitting parameters; *ε* is the PAM contraction rate; $\varepsilon =\frac{L_{0}-L}{L_{0}}$, $\;L_{0}$ is the original length of the PAM; L is the actual length of the PAM contraction; and μ is the nonlinear attenuation coefficient of shrinkage.

In this paper, we used this method to establish the dynamic model of a bionic leg with double-joint PAMs. 

### 3.2. Dynamics of Bionic Leg

The force analysis of the bionic legs is shown in Figure 4. The joint connecting rod rotates around the rotation center under the combined action of the PAM output forces $F_{p1}\;\mathrm{and}\;F_{p2}$ and the spring force $F_{spr}$. The knee joint torque and ankle joint torque are calculated as follows:(5)$$\begin{array}{c} \left\{ \begin{array}{l} \tau_{k}=F_{p1}r_{1}-F_{p2}r_{2}\\ \begin{array}{c} \tau_{a}=F_{p2}r_{3}-F_{spr}r_{4} \end{array} \end{array}\right. \end{array}$$

The Lagrange method was used to model the dynamics of the bionic legs with two joints, ignoring the influence of pneumatic artificial muscle contraction on the centroid of the connecting rod:(6)$$M(\theta )\ddot{\theta }+C(\theta ,\dot{\theta })+G(\theta )=\tau$$
where, $\theta ,\dot{\;\theta },\;\;\mathrm{and}\;\ddot{\theta }$ are the angle, angular velocity, and angular acceleration of the joints, respectively; M(θ) is the inertia matrix; $C\left(\theta ,\dot{\theta }\right)$ is the centrifugal force and Coriolis force; G(θ) is the gravity term; and τ is the torque of the ankle joint and knee joint.
(7)$$M(\theta )=\left[\begin{array}{cc} \left(m_{1}+m_{2})l_{1}{}^{2}+m_{1}l_{2}{}^{2}+2m_{2}l_{1}l_{2}\cos(\theta_{2}\right) & m_{2}l_{2}{}^{2}+m_{2}l_{1}l_{2}\cos(\theta_{2})\\ m_{2}l_{2}{}^{2}+m_{2}l_{1}l_{2}\cos(\theta_{2}) & m_{2}l_{2}(l_{1}+l_{2})\cos(\theta_{2}) \end{array}\right]$$
(8)$$C(\theta ,\dot{\theta })=\left[\begin{array}{l} C_{1}\\ C_{2} \end{array}\right]=\left[\begin{array}{c} -m_{2}l_{1}{}^{2}\sin(\theta_{2}){\dot{\theta }}_{2}{}^{2}-2m_{2}l_{1}{}^{2}\sin(\theta_{2}){\dot{\theta }}_{1}{\dot{\theta }}_{2}\\ m_{2}l_{1}{}^{2}\sin(\theta_{2}){\dot{\theta }}_{1}{}^{2} \end{array}\right]$$
(9)$$G(\theta )=\left[\begin{array}{l} G_{1}\\ G_{2} \end{array}\right]=\left[\begin{array}{c} \left(m_{1}+m_{2})gl_{1}\cos(\theta_{1})+m_{2}gl_{2}\cos(\theta_{1}+\theta_{2}\right)\\ m_{2}gl_{2}\cos(\theta_{1}+\theta_{2}) \end{array}\right]$$

The equation can be rewritten as follows:(10)$$\ddot{\theta }=-M^{-1}(\theta )C(\theta ,\dot{\theta })-M^{-1}(\theta )G(\theta )+M^{-1}\tau$$

The relationship between the bionic leg joint angle and the pneumatic muscle force is as follows:(11)$$\ddot{\theta }=-M^{-1}(\theta )(\left[\begin{array}{l} C_{1}\\ C_{2} \end{array}\right]-\left[\begin{array}{l} G_{1}\\ G_{2} \end{array}\right]+\left[\begin{array}{l} 0\\ F_{spr}r_{4} \end{array}\right]+\left[\begin{array}{cc} r_{1} & r_{2}\\ 0 & r_{3} \end{array}\right]\left[\begin{array}{l} F_{p1}\\ F_{p2} \end{array}\right])$$

## 4. Simulation of Joint Position Control with ADRC

Active disturbance rejection control does not depend on the control object model, and it performs real-time estimation and compensation for internal and external disturbances during system operation, demonstrating robustness and dynamism. The active disturbance rejection control algorithm consists of a tracking differentiator (TD), an extended state observer (ESO), and a nonlinear state error feedback (NLSEF) controller [29].

### 4.1. Joint Position Control of Bionic Leg

The position control system of the bi-joint bionic leg with ADRC is shown in Figure 5. In the ADRC process, the part outside the system control input is regarded as the “dynamic coupling” section, and the model error and external disturbances are regarded as system disturbances. Real-time observation and compensation are carried out through the extended state observer, so that the bionic leg joint coupling system is transformed into two independent joint control systems.

When $y_{i}=\theta_{i}\;\mathrm{and}\;u_{i}=p_{i}$, according to Formula (4) and Formula (11), the relationship between the joint angle and air pressure can be written as follows:(12)$$\{\begin{array}{c} {\ddot{y}}_{1}\left(t\right)=p_{1}\left[y_{1}\left(t\right).{\dot{y}}_{1}\left(t\right),y_{2}\left(t\right),{\dot{y}}_{2}\left(t\right),u_{2}\right]+w_{1}+b_{1}u_{1}\\ {\ddot{y}}_{2}\left(t\right)=p_{2}\left[y_{1}\left(t\right).{\dot{y}}_{1}\left(t\right),y_{2}\left(t\right),{\dot{y}}_{2}\left(t\right)\right]+w_{2}+b_{2}u_{2} \end{array}$$
where $b_{i}$ is the gain coefficient for control, p(∗) is the nonlinear functions of the system states and the coupling term, $w_{i}$ is the external disturbances in the loop, $u_{i}$ is the input of the system, and $y_{i}$ is the output of the system.

Taking the amount other than the system loop input as the total disturbance of the system, Equation (12) can be rewritten as:(13)$$\left\{ \begin{array}{l} {\ddot{y}}_{1}=f_{1}+b_{1}u_{1}\\ {\ddot{y}}_{2}=f_{2}+b_{2}u_{2} \end{array}\right.$$

We considered loop *i* in Formula (13) and designed an active disturbance rejection controller.

The tracking differentiator (TD) was constructed by setting the value as the input:(14)$$\left\{ \begin{array}{l} fh=fhan(v_{1,i}-v_{i},v_{2,i},r_{0,i},h_{i})\\ v_{1,i}=v_{1,i}+h_{i}v_{2,i}\\ v_{2,i}=v_{2,i}+hfh \end{array}\right.$$

According to the output signal, the extended state observer (ESO) was constructed to track and estimate the internal state and disturbance of the system in real time:(15)$$\left\{ \begin{array}{l} e=z_{1,i}-y_{i},fe=fal(e,0.5,\delta ),fe_{1}=fal(e,0.25,\delta )\\ z_{1,i}=z_{1,i}+h_{i}(z_{2,i}-\beta_{01,i}e)\\ z_{2,i}=z_{2,i}+h_{i}(z_{3,i}-\beta_{02,i}fe+b_{0,i}u_{i})\\ z_{3,i}=z_{3,i}+h_{i}\left(-\beta_{03,i}fe_{1}\right) \end{array}\right.$$

When the performance of the extended state observer is sufficient, $z_{1,i},z_{2,i},\;\;\mathrm{and}\;z_{3,i}$ can effectively estimate the state variables of the system, $x_{i,i},x_{2,i},\mathrm{and}\;x_{3,i}$, and the nonlinear state error feedback (NLSEF) is used for the feedback control of the system:(16)$$\left\{ \begin{array}{l} e_{1}=v_{1,i}-z_{1,i},e_{2}=v_{2,i}-z_{2,i}\\ u_{0}=fhan(e_{1},ce_{2},r,h_{1,i})\\ u=\frac{u_{0}-z_{3}}{b_{0,i}} \end{array}\right.$$
where, $c,r,h_{1,i}$ is the controller gain and $b_{0,i}$ is the compensation factor.

The function expressions $fhan\left(x_{1},x_{2},r,h\right)$ and fal(e,α,δ) are as follows:(17)$$fal(e,\alpha ,\delta )=\left\{ \begin{array}{l} |a{|}^{\alpha }sign(e),|e|>\delta \\ \frac{e}{\delta^{1-\alpha }},\;\;|e|\leq \delta \end{array}\right.$$
(18)$$fhan(x_{1},x_{2},r,h)\left\{ \begin{array}{l} d=rh\\ d_{0}=hd\\ y=x_{1}+hx_{2}\\ a_{1}=\sqrt{d^{2}+8r|y|}\\ a=\left\{ \begin{array}{l} x_{2}+\frac{a_{0}-d}{2}sign(y),|y|>d_{0}\\ x_{2}+\frac{y}{h}\;\;,|y|\leq d_{0} \end{array}\right.\\ fhan=-\left\{ \begin{array}{l} rsign(a),|a|>d\\ r\frac{a}{d}\;\;,|a|\leq d \end{array}\right. \end{array}\right.$$

### 4.2. Position Control Simulation of the Bionic Leg

The centroid of the bionic leg needs a certain initial velocity at the moment of departure for the leg to take off and reach a certain jump height. We interpolated the centroid motion trajectory of the bionic jumping robot in the takeoff stage using a variable quintic polynomial [30]. The joint angle was kept unchanged after leaving the ground. The corresponding joint angle variation curve was obtained from the relationship between the joint and the centroid.

A 3D model of the bionic leg was established in ADAMS software, and the controller was built using Simulink to simulate the tracking and control of the knee and ankle joint angles. The ADAMS–MATLAB co-simulation model was constructed using the co-simulation plug-in and is shown in Figure 6. ADRC and PID control were used to track the angle of the bionic leg joint, and the effects of the two control algorithms were studied.

The simulation results are shown in Figure 7 and Figure 8. Figure 7 presents the joint position tracking simulation results of the ADRC decoupling control algorithm and the PID control algorithm. Figure 8 presents the joint position error of the two control algorithms.

The simulation results show that ADRC and PID control are highly effective for double-joint angle control. The control accuracy of the ADRC decoupling control algorithm for the knee joint and ankle joint was 0.10° and 0.38° and the control accuracy of PID control for the knee joint and ankle joint was 0.51° and 1.37°, respectively. The ADRC decoupling algorithm had a better decoupling effect.

## 5. Experiment

### 5.1. Experimental System

The experimental system is shown in Figure 9. The experimental system included an air compressor; a mist separator regulator (AWM20-02BCG); a proportional pressure regulator (Festo VPPM-10L-L-1-G18-0L10H); a biarticular bionic leg model driven by PAMs (Festo pneumatic artificial muscles with lengths of 280 mm and 160 mm); and a multifunctional I/O device (NI USB-6212). The bionic leg was composed of 3D-printing resin and a carbon-fiber tube. The controller was designed using MATLAB/Simulink. The multifunctional I/O device collected the voltage signal of the angle sensor in real time, and the proportional pressure regulator controlled the output pressure of the PAMs.

### 5.2. Position Control Experiment for Biarticular Bionic Leg

The joint angle tracking control experiment was conducted according to the joint trajectory planning curve. The experimental results are shown in Figure 10 and Figure 11. Figure 10 presents the experimental results of the bionic leg joint position trajectory tracking, and Figure 11 presents the position tracking error of the bionic leg joint.

As can be seen from the angle tracking experiment, due to the equipment responses and the hysteresis and creep of the pneumatic artificial muscles, the tracking performance had a certain degree of error, and the above problems were not considered in the simulation. There was a strong tremor when switching from the take-off phase to a constant joint angle. The maximum error amplitudes of the knee joint and ankle joint angles with ADRC were 2.64° and 12.46°. The maximum tracking errors of the PID control algorithm for the knee joint and ankle joint angles were 5.14° and 20.23°. The stability time for ADRC was 0.544 s, and the stability time for the PID control algorithm was 0.83 s. The experimental results show that the ADRC decoupling control of the double-joint bionic leg position was faster and more effective than the PID control algorithm.

### 5.3. Bionic Leg Jumping Experiment

The jumping experiment results for joint angle control using ADRC are shown in Figure 12. Figure 12a displays the initial state of the bionic leg; in Figure 12b, the bionic leg joints move according to the planned trajectory; in Figure 12c, the centroid of the bionic leg reaches the maximum speed for completion of the departure, and the joint angle remains unchanged; in Figure 12d, the bionic leg reaches the highest position, and the maximum displacement in the vertical direction is 150 mm; in Figure 12e, the bionic leg begins to fall freely; and in Figure 12f, the foot of the bionic leg reaches the ground, and the horizontal displacement is 320 mm.

## 6. Discussion and Conclusions

In order to meet the requirements of agility, adaptability, and operability for jumping robots, researchers seek inspiration from animal mechanisms, imitating biological structures and actions. As an actuator with a bio-like muscle contraction process, a pneumatic artificial muscle can easily imitate biological mechanisms. The high power/mass ratio of pneumatic artificial muscles cannot be ignored. Compared with bionic legs driven by other actuators, bionic jumping legs driven by pneumatic artificial muscles can theoretically achieve a higher energy density and lighter weight. The rubber endobiliary of pneumatic artificial muscles ensures that the bionic leg has a certain degree of flexibility and impact resistance similar to that of a biological tendon to alleviate the landing. Furthermore, the safety of the gas required for pneumatic artificial muscles has certain advantages for applications in complex environments (battlefield reconnaissance, archaeological exploration, anti-terrorism operations, etc.) and human–computer interaction scenarios. However, the nonlinearity, creep, and hysteresis of pneumatic artificial muscles make their application challenging.

In this paper, inspired by the hind limbs of a jumping kangaroo, pneumatic artificial muscles were used as actuators in the design of a bionic jumping leg. The pneumatic artificial muscles were arranged according to the location of the gastrocnemius muscle in the kangaroo, and the range of ankle joint angle motion was improved by using the characteristics of double-joint muscles (transferring part of the motion to the next joint).

With the goal of coupling the bionic joints using double-joint muscles, we reduced the coupling term to the total disturbance by applying an extended state observation, so that the system was decoupled into two single-input and single-output systems, and carried out position control based on active disturbance rejection. The simulation and experiment showed that the decoupling control method based on ADRC was more effective and had a shorter adjustment time than PID control. In the experiment, the control effects decreased as a result of the hysteresis and creep effects of the pneumatic artificial muscles. We generated a jump curve using quintic variable interpolation polynomial programming. Under the control of the mechanism, the horizontal jump was 320 mm and the vertical jump was 150 mm.

## Figures and Tables

**Figure 1 micromachines-13-00827-f001:**
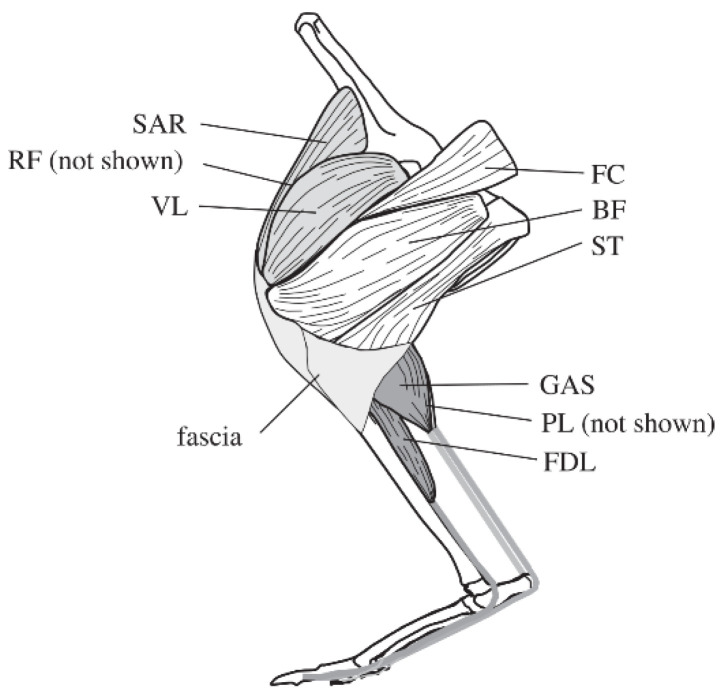
Schematic drawing of the left hind limb of a kangaroo [27].

**Figure 2 micromachines-13-00827-f002:**
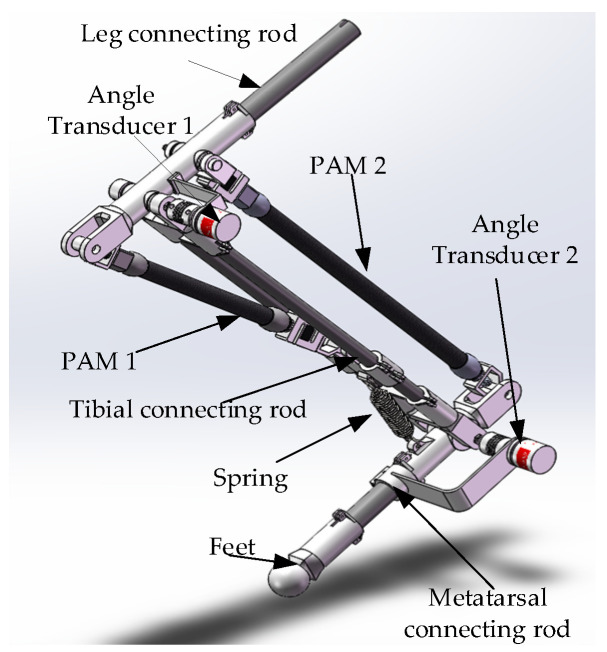
Bionic leg driven by biarticular PAM and monoarticular PAM.

**Figure 3 micromachines-13-00827-f003:**
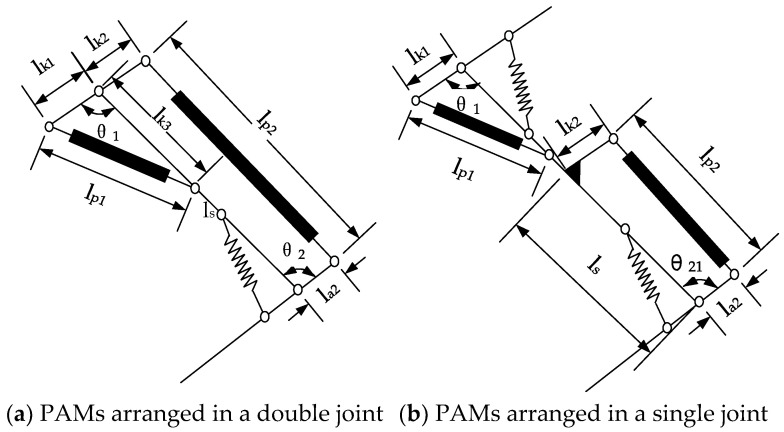
Sketch of the bionic leg structure.

**Figure 4 micromachines-13-00827-f004:**
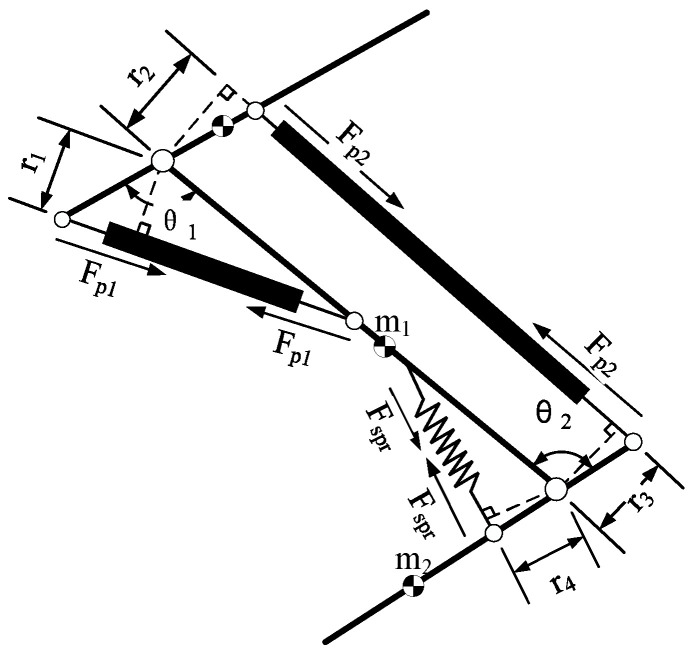
Force analysis of bionic leg.

**Figure 5 micromachines-13-00827-f005:**
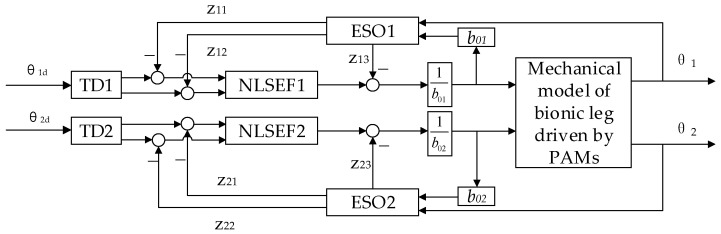
Schematic diagram of the position control of the bionic leg with ADRC.

**Figure 6 micromachines-13-00827-f006:**
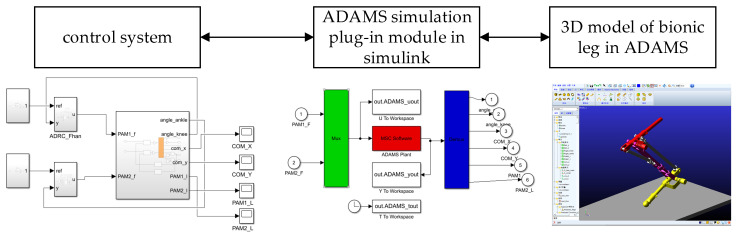
ADAMS–MATLAB co-simulation model.

**Figure 7 micromachines-13-00827-f007:**
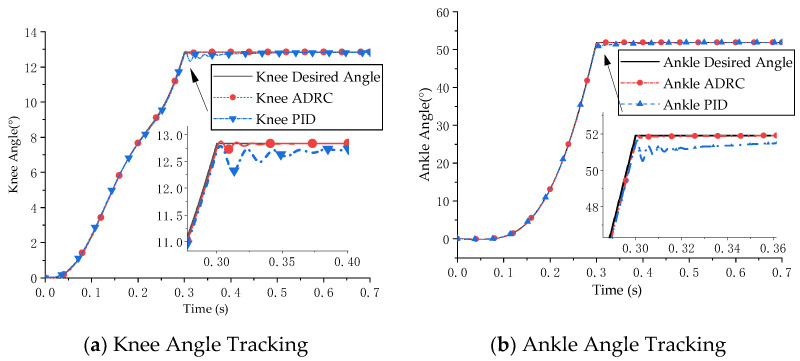
Simulation results of joint angle tracking control.

**Figure 8 micromachines-13-00827-f008:**
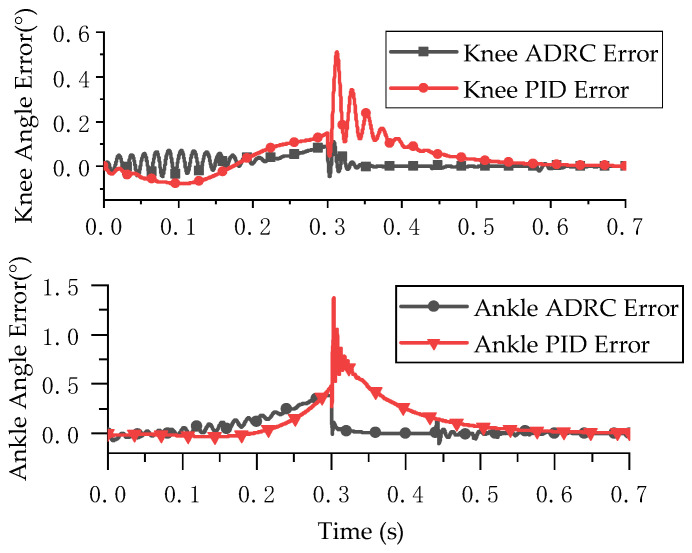
Joint angle tracking error.

**Figure 9 micromachines-13-00827-f009:**
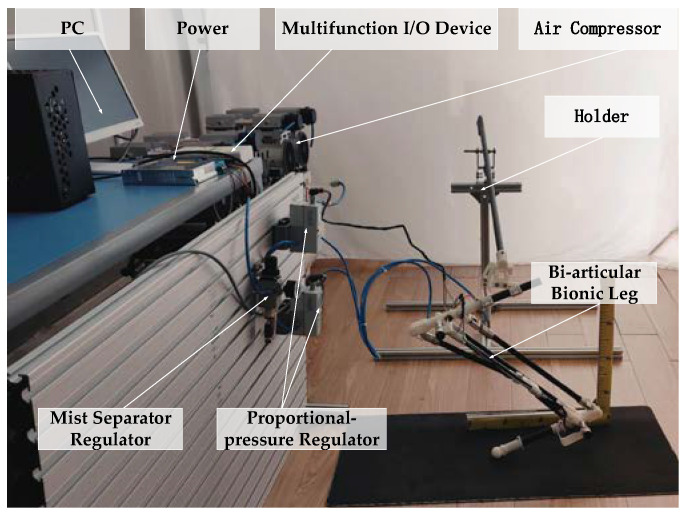
Bionic leg experimental platform.

**Figure 10 micromachines-13-00827-f010:**
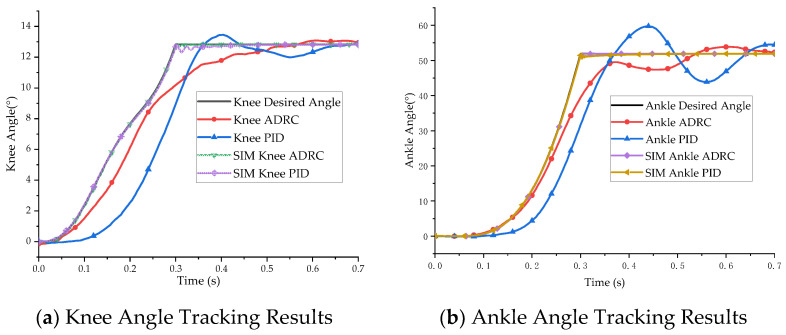
Experimental results of bionic leg joint angle control.

**Figure 11 micromachines-13-00827-f011:**
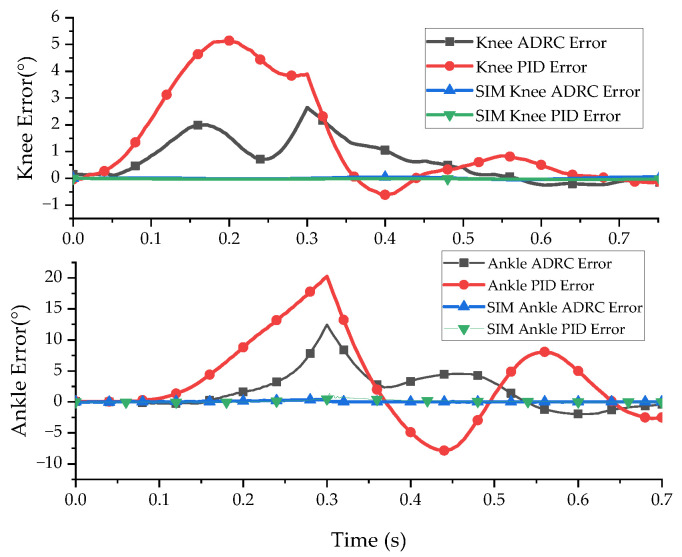
Experimental error of bionic leg joint angle control.

**Figure 12 micromachines-13-00827-f012:**
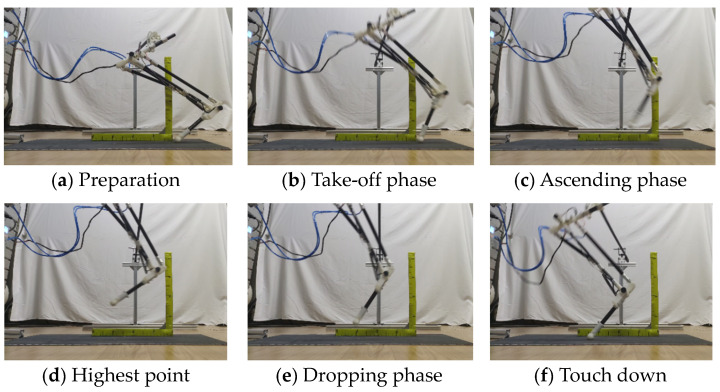
Bionic jumping leg experiment.

**Table 1 micromachines-13-00827-t001:** Bionic jumping robots driven by pneumatic artificial muscles.

Name	Mass	Bionic Prototype	Jumping Height	Highlights
Musculoskeletal quadruped robot [16]	6.0 kg	Quadruped mammals	0.254 m	Torque-Angle Relationship Control System
One-Legged Jumping Robot [17]	9.3 kg	Human	0.1 m	Antagonistic, multi-joint muscles
Athlete Robot [18]	10 kg	Human	0.5 m	Stiffness planning and soft landing
Mowgli [13]	3 kg	Human	0.4 m	Soft landing

**Table 2 micromachines-13-00827-t002:** Parameters and muscle length range of bionic leg.

Parameter	Value	Parameter	Value
*l_k_*_1_/mm	64	*l_a_*_2_/mm	55
*l_k_*_2_/mm	55	*l_p_*_1_/mm	277~307
*l_k_*_3_/mm	263	*l_p_*_2_/mm	380~433
*l_s_*/mm	438

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
