# Peer review of "Design and Joint Position Control of Bionic Jumping Leg Driven by Pneumatic Artificial Muscles"

_micromachines, 2022, doi:10.3390/mi13060827_

Round 1
Reviewer 1 Report
In this manuscript, Dai et al. developed a joint position control method for bioinspired robotic legs. The control approach is explained in detail and its performance is benchmarked against the conventional PID control in both simulation and experiments. Overall, this paper is clearly written and the figures are well plotted. The reviewer has the following comments:
- The novelty of this paper is rather unclear. In the introduction section, the authors did a literature survey, the purpose of this survey should be to introduce what has been done and what hasn't, then introduces the novelty or contribution of this paper (i.e. doing what hasn't been done). However, the literature survey in the current form is a simple summary of other peoples works. The reviewer suggests the authors to rewrite the introduction section and to define the novelties clearly.
- Also in the introduction section, the authors mentioned traditional bionic hopping robots mainly adopt motors, ... and pneumatics.... Then claim that pneumatic artificial muscles are a new type of pneumatic element. What is the difference between the 'pneumatic artificial muscle' used in this paper and other pneumatic elements in the literature? Please clearify in the revised paper.
- The authors mentioned the pneumatic artificial muscle has high power outputs. Does it really have a high power output compared to motors and hydraulics? Please provide supporting evidences.
- Please discuss the potential applications of the bionic jumping leg.
Author Response
Dear reviewer:
Thank you for your letter and the reviewers’ comments on our manuscript entitled " Research on joint position control of bionic leg jumping driven by pneumatic artificial muscles " (ID:micromachines-1709830). Those comments are very helpful for revising and improving our paper, as well as the important guiding significance to other research. We have studied the comments carefully and made corrections which we hope meet with approval. The main corrections are in the manuscript and the responds to the reviewers’ comments are as follows (the replies are highlighted in red):
Point 1: The novelty of this paper is rather unclear. In the introduction section, the authors did a literature survey, the purpose of this survey should be to introduce what has been done and what hasn't, then introduces the novelty or contribution of this paper (i.e., doing what hasn't been done). However, the literature survey in the current form is a simple summary of other people’s works. The reviewer suggests the authors to rewrite the introduction section and to define the novelties clearly.
Response 1: The introduction section has been rewritten and clearly defined novelty. Please see the attachment.
Point 2: Also in the introduction section, the authors mentioned traditional bionic hopping robots mainly adopt motors, ... and pneumatics.... Then claim that pneumatic artificial muscles are a new type of pneumatic element. What is the difference between the 'pneumatic artificial muscle' used in this paper and other pneumatic element in the literature? Please clearify in the revised paper.
Response 2: The definition of pneumatic artificial muscles and differences from other pneumatic components have been added to the manuscript. It can be found in Section 1, paragraph 2.
Point 3: The authors mentioned the pneumatic artificial muscle has high power outputs. Does it really have a high power output compared to motors and hydraulics? Please provide supporting evidences.
Response 3: The high power output evidence of pneumatic artificial muscle has been added to the manuscript. It can be found in the main points of the second paragraph of Chapter 1
Point 4: Please discuss the potential applications of the bionic jumping leg.
Response 4: This discussion has been added to the manuscript, which can be found in the first paragraph of Section 6.
Once again, thank you very much for your comments and suggestions.

Reviewer 2 Report
The work presented in this manuscript is about position control of using pneumatic artificial muscles for bionic leg jumping. The work interesting and has biomedical engineering implications. The work can be accetped for publication in this journal shall the following points be addressed.
(1) There exist huge literature and profounding reserch outputs on using pneumatic artificial muscle to actuate robotic devices. It is suggested to clearly identify the nich point of the current research with respect to literature and to identify the novelty of the work presented in the paper.
(2) In addition to use results from literature on biomimetic leg dynamics, it is suggested to incorporate measurements from human leg dynamics. From which measurements, mechanical allownce and specifications can be drived to form the model leg for investigations. At this point, no such infomration can be found in the manuscript.
(3) Active disturbance rejection control and PID control algorithms both provide good results in position control fo the bionic leg. However, these two methods are widely used. Therefore, it is not clear to this review what is the new engineering findings or advancements that the research is aiming at. Perhaps more detailed clarifications is needed.
Author Response
Dear reviewer:
Thank you for your letter and the reviewers’ comments on our manuscript entitled " Research on joint position control of bionic leg jumping driven by pneumatic artificial muscles " (ID:micromachines-1709830). Those comments are very helpful for revising and improving our paper, as well as the important guiding significance to other research. We have studied the comments carefully and made corrections which we hope meet with approval. The main corrections are in the manuscript and the responds to the reviewers’ comments are as follows (the replies are highlighted in red):
Point 1: There exist huge literature and profounding reserch outputs on using pneumatic artificial muscle to actuate robotic devices. It is suggested to clearly identify the nich point of the current research with respect to literature and to identify the novelty of the work presented in the paper.
Response 1: Inspired by the kangaroo jumping movement, this paper uses pneumatic artificial muscles as actuators to design bionic jumping legs. Among them, the pneumatic artificial muscle was arranged according to the bi-joint muscle distribution of the gastrocnemius muscle of the hind leg of the kangaroo, which alleviated the problem that the pneumatic artificial muscle limited the rotation angle of the bionic joint due to the contraction rate and improved the range of joint motion. The coupling of bi-articular muscles is estimated by using the extended state observer as the total disturbance, and the active disturbance rejection control is used for decoupling control. Detailed descriptions have been revised in the introduction section to the manuscript.
Point 2: In addition to use results from literature on biomimetic leg dynamics, it is suggested to incorporate measurements from human leg dynamics. From which measurements, mechanical allownce and specifications can be drived to form the model leg for investigations. At this point, no such infomration can be found in the manuscript.
Response 2: In this paper, the bionic jumping leg is inspired by the jumping biological kangaroo, and the bone and muscle distribution of the hind leg are designed. The reference of bionic leg structure design and the arrangement of muscle have been supplemented in the manuscript. It can be found in Section 2.1 of the manuscript.
Point 3: Active disturbance rejection control and PID control algorithms both provide good results in position control of the bionic leg. However, these two methods are widely used. Therefore, it is not clear to this review what is the new engineering findings or advancements that the research is aiming at. Perhaps more detailed clarifications is needed.
Response 3: In this paper, for the coupling term caused by bionic joint driven by bi-articular muscle, the part outside the input term is classified as the total disturbance, and the extended state observer is used to estimate and track, so as to construct the active disturbance rejection control algorithm, and then the multi-input multi-output system is decoupled into multiple single-input single-output systems for control. Then simulate and experiment. The above issues have been revised in the manuscript. They can be found in abstracts, introductions and summaries.
Once again, thank you very much for your comments and suggestions.

Reviewer 3 Report
The authors have proposed position control of bionic leg jumping driven by a pneumatic actuator in this paper. The work includes the design of the leg, dynamic modeling and simulation with experimental validation. The proposed work is thoroughly studied. However, it requires some explanation and needs more results.
- The reviewer suggests removing the "research on" word from the title. It is pretty obvious that it is research work.
- Fig 2 b is not clear how it is a single joint. Could you please elaborate better?
- The reviewer recommends combining the simulation and experimental plots. It will help in the validation of simulation and experimental results.
- Please add a video of the simulation and experimental of the jumping leg.
- There are several bionic jumping legs. It would be better if the authors could add a comparative table to see where the proposed bionic leg robot stands in terms of its performance.
Author Response
Point 1: The reviewer suggests removing the "research on" word from the title. It is pretty obvious that it is research work.
Response 1: The title name has been modified to ‘Deign and Joint Position Control of Bionic Jumping Leg Driven by Pneumatic Artificial Muscles’.
Point 2: Fig 2 b is not clear how it is a single joint. Could you please elaborate better?
Response 2: Monoarticular muscles refer to the muscle acting on a single joint, contraction only affects the movement of a single joint. In order to make a comparative analysis, in the case of keeping the parameters unchanged, the influence of a segment of the double joint muscle on a joint will be removed to make it a single joint muscle, that is, the fixed point, does not actually rotate, while ensuring joint rotation. In order to make the expression more clear, the pictures in the manuscript have been centrally revised. The picture has been revised to Fig. 3b.
Point 3: The reviewer recommends combining the simulation and experimental plots. It will help in the validation of simulation and experimental results.
Response 3: The simulation and experimental plots have been combined and the manuscript pictures have been modified. The picture has been revised to Fig. 10. and Fig. 11.
Point 4: Please add a video of the simulation and experimental of the jumping leg.
Response 4: The video of the simulation and experimental of the jumping leg has been submitted to the editor by mail.
Point 5: There are several bionic jumping legs. It would be better if the authors could add a comparative table to see where the proposed bionic leg robot stands in terms of its performance.
Response 5: Tables on performance and parameters of bionic legs driven by pneumatic artificial muscles have been added to the manuscript. It can be found in Table 1 of the Introduction

Round 2
Reviewer 2 Report
Thank you authors for the quick and thorough responses to my review comments. The videos and the responses are very much appreciated and have answered my previsous questions and comments. I think it is now in a good shape for possible publication in this jouranl.